# ACTIVATION RELAXATION: A LOCAL DYNAMICAL APPROXIMATION TO BACKPROP IN THE BRAIN

## ABSTRACT

The backpropagation of error algorithm (backprop) has been instrumental in the recent success of deep learning. However, a key question remains as to whether backprop can be formulated in a manner suitable for implementation in neural circuitry. The primary challenge is to ensure that any candidate formulation uses only local information, rather than relying on global signals as in standard backprop. Recently several algorithms for approximating backprop using only local signals have been proposed. However, these algorithms typically impose other requirements which challenge biological plausibility: for example, requiring complex and precise connectivity schemes, or multiple sequential backwards phases with information being stored across phases. Here, we propose a novel algorithm, Activation Relaxation (AR), which is motivated by constructing the backpropagation gradient as the equilibrium point of a dynamical system. Our algorithm converges rapidly and robustly to the correct backpropagation gradients, requires only a single type of computational unit, utilises only a single parallel backwards relaxation phase, and can operate on arbitrary computation graphs. We illustrate these properties by training deep neural networks on visual classification tasks, and describe simplifications to the algorithm which remove further obstacles to neurobiological implementation (for example, the weight-transport problem, and the use of nonlinear derivatives), while preserving performance.

In the last decade, deep artificial neural networks trained through the backpropagation of error algorithm (backprop) (Werbos, 1982; Griewank et al., 1989; Linnainmaa, 1970) have achieved substantial success on a wide range of difficult tasks such as computer vision and object recognition (Krizhevsky et al., 2012; He et al., 2016), language modelling (Vaswani et al., 2017; Radford et al., 2019; Brown et al., 2020), unsupervised representation learning (Radford et al., 2015; Oord et al., 2018), image and audio generation (Goodfellow et al., 2014; Salimans et al., 2017; Jing et al., 2019; Oord et al., 2016; Dhariwal et al., 2020) and reinforcement learning (Silver et al., 2017; Mnih et al., 2015; Schulman et al., 2017; Schrittwieser et al., 2019). The impressive performance of backprop is due to the fact that it precisely computes the sensitivity of each parameter to the output (Lillicrap et al., 2020), thus solving the credit assignment problem which is the task of determining the individual contribution of each parameter (potentially one of billions in a deep neural network) to the global outcome. Given the correct credit assignments, network parameters can be straightforwardly, and independently, updated in the direction which maximally reduces the global loss. The brain also faces a formidable credit assignment problem – it must adjust trillions of synaptic weights, which may be physically and temporally distant from their global output, in order to improve performance on downstream tasks [1]. Given that backprop provides a successful solution to this problem (Baldi & Sadowski, 2016), a large body of work has investigated whether synaptic plasticity in the brain could be interpreted as implementing or approximating backprop (Whittington & Bogacz, 2019; Lillicrap et al., 2020). Recently, this idea has been buttressed by findings that the representations learnt by backprop align closely with representations extracted from cortical neuroimaging data (Cadieu et al., 2014; Kriegeskorte, 2015).

Due to the nonlocality of its learning rules, a direct term-for-term implementation of backprop is likely biologically implausible (Crick, 1989). It is important to note that biological-plausibility is an

---

[1]It is unlikely that the brain optimizes a single cost function, as assumed here. However, even if functionally segregated areas can be thought of as optimising some combination of cost functions, the core problem of credit assignment remains.

amorphous term in the literature, with many possible readings. Here, we specifically mean that the algorithm requires only local information to be present at each synapse, information is processed according to straightforward linear, or Hebbian-like update rules. It is important to note that here (as in much of the literature), we use the abstraction of rate-coded leaky-integrate-and-fire neurons. The extension to more neurophysiologically grounded spiking models is an exciting area of future work. In recent years, there has been a substantial amount of work developing more biologically plausible approximations that rely solely on local information. (Whittington & Bogacz, 2017; Bengio & Fischer, 2015; Bengio et al., 2017; Scellier et al., 2018a;b; Sacramento et al., 2018; Guerguiev et al., 2017; Lee et al., 2015; Lillicrap et al., 2014; Nøkland, 2016; Millidge et al., 2020a; Ororbia II et al., 2017; Ororbia & Mali, 2019; Ororbia et al., 2020). The local learning constraint requires that the plasticity at each synapse depend only on information which is (in-principle) locally available at that synapse, such as pre- and post- synaptic activity in addition to local point derivatives and potentially the weight of the synaptic connection itself. The recently proposed NGRAD hypothesis (Lillicrap et al., 2020) offers a unifying view on many of these algorithms by arguing that they all approximate backprop in a similar manner. Specifically, it suggests that all of these algorithms approximate backprop by implicitly representing the gradients in terms of neural activity differences, either spatially between functionally distinct populations of neurons or neuron compartments (Whittington & Bogacz, 2017; Millidge et al., 2020a), or temporally between different phases of network operation (Scellier & Bengio, 2017; Scellier et al., 2018b).

Another way to understand local approximations to backprop comes from the notion of the learning channel (Baldi & Sadowski, 2016). In short, as the optimal parameters must depend on the global outcomes or targets, any successful learning rule must propagate information backwards from the targets to each individual parameter in the network that contributed to the outcome. We argue that there are two primary ways to achieve this[2]. First, a sequential backwards pass could be used, where higher layers propagate information about the targets backwards in a layer-wise fashion, such that the gradients of each layer are simply a function of the layer above. This is the approach employed by backprop, which propagates error derivatives explicitly. Other algorithms such as target-propagation (Lee et al., 2015) also perform a backwards pass using only local information, but do not asymptotically approximate the backprop updates. Secondly, instead of a sequential backwards pass, information could be propagated through a dynamical relaxation underwritten by recurrent dynamics, such that information about the targets slowly 'leaks' backwards through the network over the course of multiple dynamical iterations. Examples of such dynamical algorithms include predictive coding (Whittington & Bogacz, 2017; Friston, 2003; 2005; Millidge, 2019) and contrastive Hebbian methods such as equilibrium-prop (Xie & Seung, 2003; Scellier & Bengio, 2017),which both asymptotically approximate the error gradients of backprop using only local learning rules.

To our knowledge, all of the algorithms in the literature, including those mentioned above, have utilised implicit or explicit activity differences to represent the necessary derivatives, in line with the NGRAD hypothesis. Here, we derive an algorithm which converges to the exact backprop gradients without utilising any layer-wise notion of activity differences. Our algorithm, which we call Activation Relaxation (AR), is derived by postulating a dynamical relaxation phase in which the neural activities trace out a dynamical system which is designed to converge to the backprop gradients. The resulting dynamics are extremely simple, and require only local information, meaning that they could, in principle be implemented in neural circuitry. Unlike algorithms such as predictive coding, AR only utilises a single type of neuron (instead of two separate populations – one encoding values and one encoding errors), and unlike contrastive Hebbian methods it does not require multiple distinct backwards phases (only a feedforward sweep and a relaxation phase). This simplification is beneficial from the standpoint of neural realism, as in the case of predictive coding, there is little evidence for the presence of specialised prediction-error neurons throughout cortex (Walsh et al., 2020) [3]. Unlike contrastive Hebbian methods, AR does not require the coordination and storage of information across multiple backwards phases, which would pose a substantial challenge for decentralised neural circuitry.

---

[2]A third method is to propagate information about the targets through a global neuromodulatory signal which affects all neurons equally. However because this does not provide precise vector feedback, the implicit gradients computed have extremely high variance, typically leading to slow and unstable learning (Lillicrap et al., 2020).

[3]There is, of course, substantial evidence for dopaminergic reward-prediction error neurons in midbrain areas (Bayer & Glimcher, 2005; Glimcher, 2011; Schultz, 1998; Schultz & Dickinson, 2000)

We empirically demonstrate that the AR algorithm accurately approximates the backprop gradients and can be used to successfully train deep neural networks on the MNIST and FashionMNIST tasks, where we demonstrate performance directly equivalent to backprop. Finally, we show that several of the remaining biologically implausible aspects of the algorithm, such as 'weight-transport' (Crick, 1989), and the evaluation of nonlinear derivatives, can be removed, resulting in an updated form of AR that requires extremely simple connectivity patterns and plasticity rules. Importantly, we show that this simplified algorithm can still train deep neural networks to high levels of performance, despite no longer approximating exact backprop gradients.

# 1 METHODS

We consider the simple case of a fully-connected deep multi-layer perceptron (MLP) composed of $L$ layers of rate-coded neurons trained in a supervised setting[4]. The firing rates of these neurons are represented as a single scalar value $x_i^l$, referred to as the neurons activation, and a vector of all activations at given layer is denoted as $x^l$. The activation's of the hierarchically superordinate layer are a function of the hierarchically subordinate layer's activations $x^{l+1} = f(W^l x^l)$, where $W^l \in \Theta$ is the set of synaptic weights, and the product of activation and weights is transformed through a nonlinear activation function $f$. The final output $x^L$ of the network is compared with the desired targets $T$, according to some loss function $\mathcal{L}(x^L, T)$. In this work, we take this loss function to be the mean-squared-error (MSE) $\mathcal{L}(x^L, T) = \frac{1}{2}\sum_i (x_i^L - T_i)^2$, although the algorithm applies to any other loss function without loss of generality (see Appendix B). We denote the gradient of the loss with respect to the output layer as $\frac{dL}{dx^L}$. In the case of the MSE loss, the gradient of the output layer is just the prediction error $\epsilon^L = (x^L - T)$. Backprop proceeds by computing the gradient of the loss function with respect to the weights $W^l$ using the chain rule,

$$\frac{\partial L}{\partial W^l} = \frac{\partial L}{\partial x^{l+1}} \frac{\partial x^{l+1}}{\partial W^l}$$
$$= \frac{\partial L}{\partial x^{l+1}} f'(W^l x^l) x^{L^T} \tag{1}$$

where $f'$ denotes the derivative of the activation function. The difficulty lies in computing the $\frac{\partial L}{\partial x^{l+1}}$ term using only local information. This is achieved by repeatedly applying the chain rule $\frac{\partial L}{\partial x^l} = \frac{\partial L}{\partial x^{l+1}} \frac{\partial x^{l+1}}{\partial x^l}$, allowing the derivatives to be computed recursively from the output loss. However, this procedure is not local since the update at each layer depends on the gradients of all superordinate layers in the hierarchy. Here, we propose a method for computing the activation derivatives, $\frac{\partial L}{\partial x^l}$, using a dynamical systems approach. Specifically, we define a dynamical system where the activations of each node at equilibrium correspond to the backpropagated gradients. Note that this is in contrast with more typical approaches where a dynamical system is designed to converge to the minimum of some global loss. The simplest system to achieve this is a leaky-integrator driven by top-down feedback,

$$\frac{dx^l}{dt} = -x^l + \frac{\partial L}{\partial x^l} \tag{2}$$

which, at equilibrium, converges to

$$\frac{dx^l}{dt} = 0 \implies x^{*l} = \frac{\partial L}{\partial x^l}$$

Furthermore, by the chain rule we can write Equation 2 as,

$$\frac{dx^l}{dt} = -x^l + \frac{\partial L}{\partial x^{l+1}} \frac{\partial x^{l+1}}{\partial x^l}\bigg|_{x^l = \bar{x}^l}$$

where $\bar{x}^l$ is the value of $x^l$ computed in the forward pass. We can express this in terms of the equilibrium activation of the superordinate layer,

$$\frac{dx^l}{dt} = -x^l + x^{*l+1} \frac{\partial x^{l+1}}{\partial x^l}\bigg|_{x^l = \bar{x}^l} \tag{3}$$

---

[4]Extensions to other architectures are relatively straightforward and will be investigated in future work. In appendix A we show that the approach can be extended to arbitrary directed acyclic graphs (DAGs), which encompasses all standard machine learning architectures such as CNNs, LSTMs, ResNets, transformers, etc.

To achieve these dynamics exactly in a multilayered network would require the sequential convergence of layers, as each layer must converge to equilibrium before the dynamics of the layer below can operate. However, to enable updates across multiple layers simultaneously, we approximate the equilibrium activation of the layer above with the layer's current activation, yielding,

$$\frac{dx^l}{dt} = -x^l + x^{*l+1} \frac{\partial x^{l+1}}{\partial x^l} \Big|_{x^l = \bar{x}^l}$$

$$\approx -x^l + x^{l+1} \frac{\partial x^{l+1}}{\partial x^l} \Big|_{x^l = \bar{x}^l} \tag{4}$$

$$\approx -x^l + x^{l+1} f'(W^l, \bar{x}^l) W^{l^T} \tag{5}$$

Despite this approximation, we argue that the system nevertheless converges to the same optimum as Equation 3. Specifically, because we evaluate $\frac{\partial x^{l+1}}{\partial x^l}$ at the feedforward pass value $\bar{x}^l$, this term remains constant throughout the relaxation phase [5]. Keeping this term fixed effectively decouples the each layer from any bottom-up influence. If the top-down input is also constant, because it has already converged so that $x^{l+1} \approx x^{l+1*}$, then the dynamics become linear, and the system is globally stable due to possessing a Jacobian which is everywhere negative-definite. The top-layer is provided with the stipulatively correct gradient, so it must converge. Recursing backwards through each layer, we see that once the top-level has converged, so too must the penultimate layer, and so through to all layers. Although this argument is somewhat heuristic, in section 2 we provide empirical results showing that it rapidly and robustly converges to the exact numerical gradients in practice.

Equation 4 forms the backbone of the activation-relaxation (AR) algorithm. The algorithm proceeds as follows. First, a standard forward pass computes the network output, which is compared with the target to calculate the top-layer error derivative $\epsilon_L$ and thus update the activation of the penultimate layer. [6]. Then, the network enters into a relaxation phase where Equation 4 is iterated globally for all layers until convergence for each layer. Upon convergence, the activations of each layer are precisely equal the backpropagated derivatives, and are used to update the weights (via Equation 1).

## 1.1 RELATED WORK

There has been a substantial amount of work addressing the challenge of developing biologically plausible implementations of, or approximations to, backprop, with numerous schemes now available in the literature. Many attempts have been made to address or work around the weight transport problem – the requirement that backwards information be conveyed by the transpose of the forward weights – by either simply using random backwards weights (Lillicrap et al., 2016), directly transmitting gradients backwards to all layers from the output layer (Nøkland, 2016), or simply learning the backwards weights themselves (Amit, 2019; Akrout et al., 2019). In addition, recurrent algorithms have been developed that can converge to representations of the backprop gradients with only local rules in an iterative fashion. These algorithms include predictive coding (Whittington & Bogacz, 2017; Millidge, 2019; Millidge et al., 2020a; Friston, 2005), where gradients are implicitly computed by minimizing layerwise prediction errors, equilibrium-prop (Scellier & Bengio, 2017; Scellier et al., 2018b), which uses a constrastive Hebbian learning approach where gradients are computed through the differences between a free and a fixed phase, and target-propagation (Lee et al., 2015), in which layers are optimized to minimize a layerwise target, and the local-representation-alignment (LRA) family of algorithms (Ororbia II et al., 2017; Ororbia et al., 2018; Ororbia & Mali, 2019) which is similar to target-prop except that the targets it computes induce the local layer-wise target minimization to encourage each layer to produce representations which will aid the layer above.

AR differs from equilibrium-prop (EP) in that it only requires one iterative phase (the relaxation phase), and the values of the neurons in the activation phase represent the gradients instead of the difference between the activities in each phase, as in EP. Similarly, AR differs from predictive coding in that it does not require a separate population of error units, but instead represents the gradients

---

[5]The need to keep this term fixed throughout the relaxation phase does present a potential issue of biological plausibility. In theory it could be maintained by short-term synaptic traces, and for some activation functions such as rectified linear units it is trivial. Moreover, later we show that this term can be dropped from the equations without apparent ill-effect

[6]This top-layer error is simply a prediction error for the MSE loss, but may be more complicated and less biologically-plausible for arbitrary loss functions

---

**Algorithm 1:** Activation Relaxation

---

**Data:** Dataset $\mathcal{D} = \{\mathbf{X}, \mathbf{T}\}$, parameters $\Theta = \{W^0 \ldots W^L\}$, inference learning rate $\eta_x$, weight
     learning rate $\eta_\theta$.

```
/* Iterate over dataset                                                    */
```
**for** $(x^0, t \in \mathcal{D})$ **do**
```
    /* Initial feedforward sweep                                           */
```
    **for** $(x^l, W^l)$ *for each layer* **do**
        $x^{l+1} = f(W^l, x^l)$
```
    /* Begin backwards relaxation                                          */
```
    **while** *not converged* **do**
```
        /* Compute final output error                                      */
```
        $\epsilon^L = T - x^L$
        $dx^L = -x^L + \epsilon^L \frac{\partial \epsilon^L}{\partial x^L}$
        **for** $x^l, W^l, x^{l+1}$ *for each layer* **do**
```
            /* Activation update                                           */
```
            $dx^l = -x^l + x^{l+1}\frac{\partial x^{l+1}}{\partial x^l}$
            $x^{l^{t+1}} \leftarrow x^{l^t} + \eta_x dx^l$
```
    /* Update weights at equilibrium                                       */
```
    **for** $W^l \in \{W^0 \ldots W^L\}$ **do**
        $W^{l^{t+1}} \leftarrow W^{l^t} + \eta_\theta x^l \frac{\partial x^l}{\partial W^l}$

---

directly in the activities of the neurons during the relaxation phase, rather than through prediction errors. The iterative form of LRA proposed in (Ororbia et al., 2018) is perhaps most similar to the AR algorithm, but significant differences still remain. AR is derived straightforwardly from a dynamical systems perspective on approximating the backprop gradient, while LRA is based on a variant of target-propagation. More importantly, AR directly optimizes the post-activations using the top-down information in the relaxation phase whereas iterative LRA optimizes the pre-activations before they are passed through the nonlinear activation function against the discrepancy between target and neural output. Due to this nonlinearity, the target-discrepancy does not exactly correspond to the backpropagated gradient in LRA, and hence the overall updates do not converge to backprop.

## 2 RESULTS

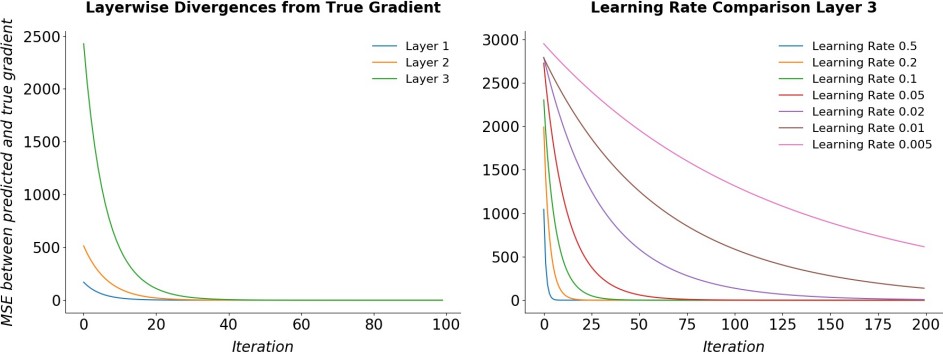

Figure 1: Mean square error between AR and the exact backpropagated gradients on a 3 layer MLP. Left: convergence each layer. Right: the effect of the learning rate on the rate of convergence.

We first demonstrate that our algorithm can train a deep neural network with equal performance to backprop. For training, we utilised the MNIST and Fashion-MNIST (Xiao et al., 2017) datasets. The MNIST dataset consists of 60000 training and 10000 test 28x28 images of handwritten digits, while the Fashion-MNIST dataset consists of 60000 training and 10000 test 28x28 images of clothing items. The Fashion-MNIST dataset is designed to be identical in shape and size to MNIST while being

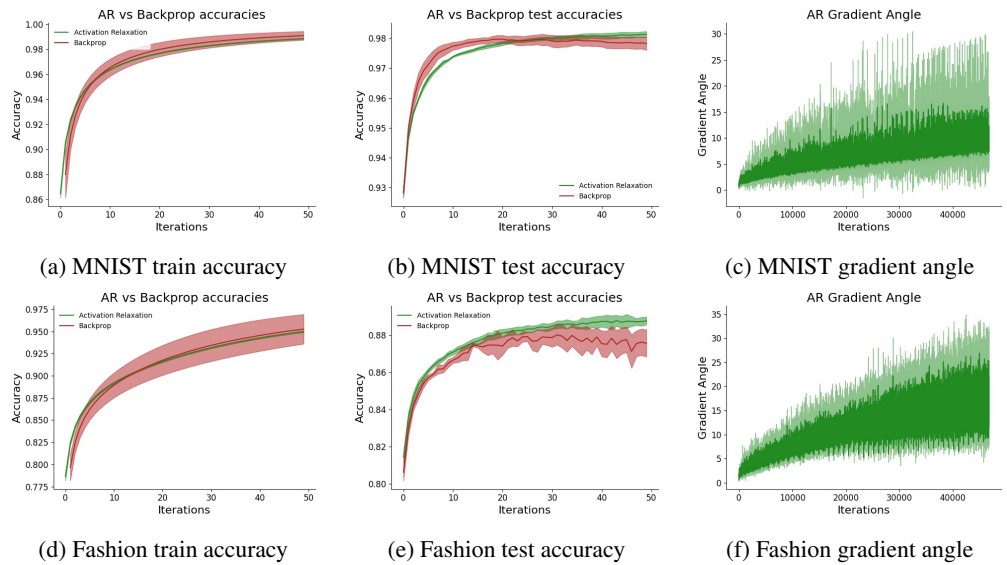

Figure 2: Train and test accuracy and gradient angle (cosine similarity) for AR vs backprop for MNIST and Fashion-MNIST datasets.

harder to solve. We used a 4-layer fully-connected multi-layer perceptron (MLP) with rectified-linear activation functions and a linear output layer. The layers consisted of 300, 300, 100, and 10 neurons respectively. In the dynamical relaxation phase, we integrate Equation 5 with a simple first-order Euler integration scheme. $x^{l^{t+1}} = x^{l^t} - \eta_x \frac{dx^l}{dt}$ where $\eta_x$ was a learning rate which was set to $0.1$. The relaxation phase lasted for 100 iterations, which we found sufficient to closely approximate the numerical backprop gradients. After the relaxation phase was complete, the weights were updated using the standard stochastic gradient descent opteimizer, with a learning rate of 0.0005. The weights were initialized as draws from a Gaussian distribution with a mean of 0 and a variance of 0.05. Hyperparameter values were chosen based off initial intuition and were not found using a grid-search. A full table of hyperparameter values is presented in Apppendix C. The AR algorithm was applied to each minibatch of 64 digits sequentially. The network was trained with the mean-squared-error loss.

Figure 1 shows that during the relaxation phase the activations converge precisely to the gradients obtained by backpropagating backwards through the computational graph of the MLP. In Figure 2 we show that the training and test performance of the network trained with activation-relaxation is nearly identical to that of the network trained with backpropagation, thus demonstrating that our algorithm can correctly perform credit assignment in deep neural networks with only local learning rules. We also empirically investigate the angle between the AR-computed gradient updates and the true backpropagated updates. The gradient angle $\mathcal{A}$ was computed using the cosine similarity metric $\mathcal{A}(\nabla_\theta) = \cos^{-1} \frac{\nabla_\theta^T \nabla_\theta^*}{||\nabla_\theta|| ||\nabla_\theta^*||}$, where $\nabla_\theta$ was the AR-computed gradients and $\nabla_\theta^*$ were the backprop gradients. To handle the fact that we had gradient matrices while the cosine similarity metric only applies to vectors, following (Lillicrap et al., 2016), we simply flattened the gradient matrices into vectors before performing the computation. We see that the updates computed by AR are very close in angle to the backprop updates (within under 10 degrees), although the angle increases slightly over the course of training. The convergence in training and test accuracies between the AR and backprop shows that this slight difference in gradient angle is not enough to impede effective credit assignment and learning in AR. In Appendix C, we take a step towards demonstrating the scalability of this algorithm, by showing preliminary results that indicate that AR, including with the biologically plausible simplifications introduced below, can scale to deeper CNN architectures and more challenging classification tasks.

## 3 REMOVING CONSTRAINTS

### 3.1 WEIGHT TRANSPORT

Although the AR algorithm only utilises local learning rules to approximate backpropagation, there are still some remaining biological implausibilities in the algorithm. Following (Millidge et al., 2020c), we show how simple modifications to the algorithm can remove these implausible constraints on the algorithm while retaining high performance. The most pressing difficulty is the weight-transport problem (Crick, 1989), which concerns the $\theta^T$ term in Equation 5. In effect, the update rule for the activations during the relaxation phase consists of the activity of the layer above propagated backwards through the transpose of the forwards weights. However, in a neural circuit this would require either being able to transmit information both forwards and backwards symmetrically across synapses, which has generally been held to be implausible, or else that the brain must maintain an identical copy of 'backwards weights' which are kept in sync with the forwards weights during learning. However, in recent years, there has been much work showing that the precise equality of forward and backwards weights that underlies the weight transport problem is simply not required. Surprisingly, Lillicrap et al. (2016) showed that fixed *random* feedback weights suffice for learning in simple MLP networks, as the forward weights learn to align with the fixed backwards weights to convey useful feedback signals. Later work (Amit, 2019; Akrout et al., 2019; Ororbia & Mali, 2019) has shown that it is also possible and more effective to learn the backwards weights from a random initialisation, where learning can take place with a local and Hebbian learning rule. Feedback alignment replaces the $\theta^T$ in Equation 5 with fixed random backwards weights $\psi$. The updated Equation 5 reads,

$$\frac{dx^l}{dt} = x^l - x^{l+1} f'(W^l x^l) \psi^l$$

The (initially random) backwards weights can also be trained with the local and Hebbian learning rule which is a simple Hebbian update between the activations of the layer and the layer above.

$$\frac{d\psi^l}{dt} = x^{l+1^T} f'(W^l x^l) x^l \tag{6}$$

The backwards weights were initialized as draws from a 0 mean, 0.05 variance Gaussian. In Figure 3.a) we show that strong performance is obtained with the learnt backwards weights. We found that using random feedback weights without learning (i.e. feedback alignment), typically converged to a lower accuracy and had a tendency to diverge, which may be due to a simple Gaussian weight initialization used here. Nevertheless, when the backwards weights are learnt, we find that the algorithm is stable and can obtain performance comparable with using the exact weight transposes. This approach of using learnable backwards weights to solve the weight transport problem has been similarly investigated in (Amit, 2019; Akrout et al., 2019), however these papers simply implement backprop with the learnt backwards weights. Here we show that learnable backwards weights can be combined with a fundamentally local learning rule, while maintaining training performance. In Figure 4.a) we plot the angle between the AR with learnable backwards weights and the true BP gradients. The angle starts out very large (about 70 degrees) since the backwards weights are randomly initialized but then rapidly decreases to about 30 degrees as the backwards weights are learnt, which seems empirically to be sufficient to enable strong learning performance.

### 3.2 NONLINEARITY DERIVATIVES

The second potential biological implausibility in the algorithm is the requirement to multiply the weight and activation updates with the derivative of the activation function. While in theory this only requires a derivative to be calculated locally for each neuron, whether biological neurons could compute this derivative is an open question. We experiment with simply removing the nonlinear derivatives in question from the update so that the updated Equation 5 now simply reads,

$$\frac{dx^l}{dt} = x^l - x^{l+1} W^{l^T} \tag{7}$$

Although the gradients are no longer match backprop, we show in Figure 3.b) that learning performance against the standard model is relatively unaffected, showing that the influence of the nonlinear derivative is small. We hypothesise that by removing the nonlinear derivative, we are effectively projecting the backprop update onto the closest linear subspace, which is still sufficiently close in

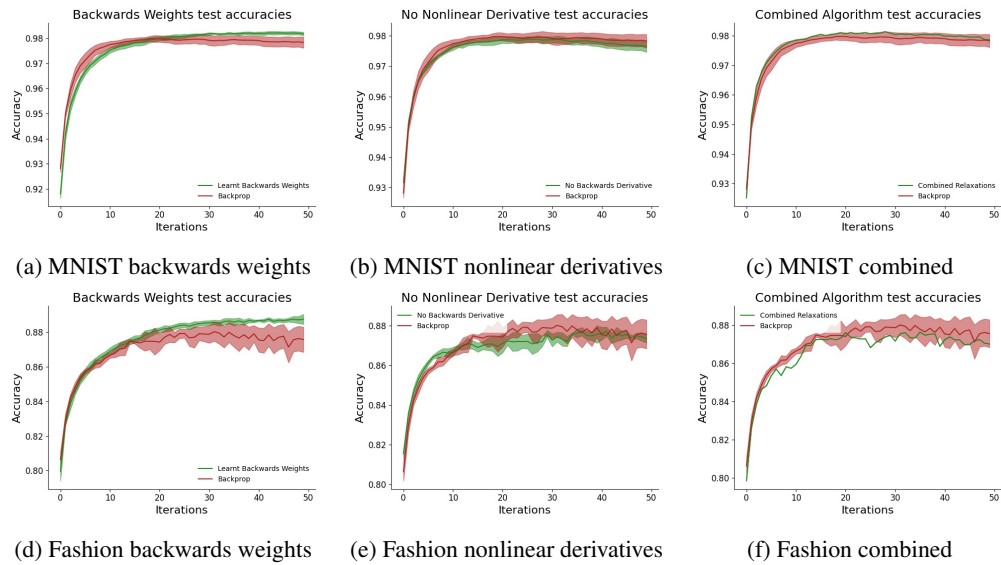

(a) MNIST backwards weights    (b) MNIST nonlinear derivatives    (c) MNIST combined

(d) Fashion backwards weights    (e) Fashion nonlinear derivatives    (f) Fashion combined

Figure 3: Train and test accuracy and gradient angle (cosine similarity) for AR vs backprop for MNIST and Fashion-MNIST datasets.

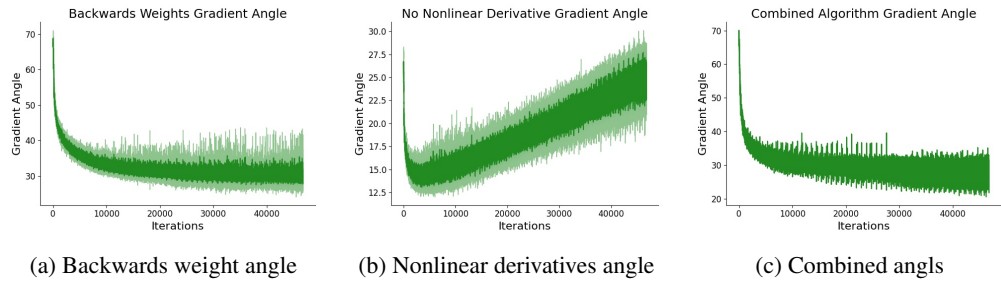

(a) Backwards weight angle    (b) Nonlinear derivatives angle    (c) Combined angls

Figure 4: Angle between the AR and backprop updates in the learnable backwards weights, no nonlinear derivatives, and the combined conditions.

angle to the true gradient that it can support learning. By explicitly plotting the angle (Figure 4.b), we see that it always remains under about 30 degrees, sufficient for learning, although the angle appears to rise over the course of training, potentially due to the gradients becoming smaller and more noisy as the network gets closer to convergence.

Moreover, we can *combine* these two changes of the algorithm such that there is both no nonlinear derivative and also learnable backwards weights. Perhaps surprisingly, when we do this we retain equivalent performance to the full AR algorithm (see Figure 3.c), and therefore a valid approximation to backprop in an extremely simple and biologically plausible form. The activation update equation for the fully simplified algorithm is:

$$\frac{dx^l}{dt} = x^l - x^{l+1}\psi \tag{8}$$

which requires only locally available information and is mathematically very simple. In effect, each layer is only updated using its own activations and the activations of the layer above mapped backwards through the feedback connections, which are themselves learned through a local and Hebbian learning rule. This rule maintains high training performance and a close angle between its updates and the true backprop updates (Figure 4.c), and is, at least in theory, relatively straightforward to implement in neural or neuromorphic circuitry.

## 4 DISCUSSION

We have shown that by taking a relatively novel perspective on the problem of approximating backprop through recurrent dynamics – by explicitly designing a dynamical system to converge on

the gradients we wish to approximate – one can straightforwardly derive from first principles an extremely simple algorithm which asymptotically approximates backprop using only local learning rules. Our algorithm requires only a feedforward pass and then a dynamical relaxation phase which we show empirically converges quickly and robustly to the exact numerical gradients computed by backprop. We demonstrate that our algorithm can be used to train deep MLPs to obtain identical performance to backprop. We then show that two key remaining biological implausibilities of the algorithm – the weight transport problem, and the need for nonlinear derivatives – can be removed without apparent harm to performance. The weight transport problem can seemingly be solved by postulating a second set of independent backwards weights which can be trained in parallel with the forward weights with a Hebbian update rule. While we have found that the nonlinear derivatives can be simply dropped from the learning rules, and does not appear to significantly harm learning performance. Future work should test whether performance is maintained on more challenging tasks. When these adjustments to the algorithm are combined, we obtain the simple and elegant update rule shown in Equation 7.

The AR algorithm does away with much of the complexity of competing schemes. Unlike contrastive Hebbian approaches, it does not require the storage of information across distinct backwards dynamical phases such as the 'free phase' and the 'clamped phase'. Unlike predictive coding approaches (Whittington & Bogacz, 2017), error-dendrite methods (Sacramento et al., 2018), or 'ghost units' (Mesnard et al., 2019), AR does not require multiple distinct neural populations with separate update rules and dynamics. Instead, the connectivity scheme in AR is identical to that of a standard MLP where only a single type of neuron is required. Since it can approximate backprop without any implicit or explicit representation of activity differences across time or space, the AR algorithm thus provides a counterexample to the NGRAD hypothesis, suggesting a more nuanced definition is required. Although the gradients themselves are not represented through an activity or temporal difference, the update rules (Equation 5 and 7) can be interpreted as a prediction error between the current activity and the activity of the layer above mapped through the backwards weights. This rule is strongly reminiscent of the target-prop updates, suggesting some points of commonality which would be interesting to investigate in future work. However, the NGRAD hypothesis as stated in (Lillicrap et al., 2020) requires that gradients be encoded directly in spatial or temporal differences. Our method shows that this is not necessary and that instead inter-layer errors can be used to drive dynamics which converge to the correct gradients.

Although the AR algorithm (especially the simplified version) takes a strong step towards biological realism, it still possesses several weaknesses which may render a naive implementation in neural circuitry challenging. The primary difficulty is the necessity of two distinct phases – a feedforward phase which sets the activations directly by bottom-up connectivity, and a dynamical relaxation phase where the activations are then adjusted using local learning rules. In effect, AR uses the 'activation' units for two distinct purposes – to represent both the activations and their gradients – at different times. While this is not a problem in the current paradigm where i.i.d items are presented sequentially to the network, for a biological brain enmeshed in continuous-time sensory exchange, the requirement for a dynamical relaxation adjusting the firing rates of all neurons, before the next sensory stimulus can be processed would be a serious challenge. By reusing the activations for two different tasks, the brain must either rigidly stick to two separate phases – an 'inference' phase and a 'learning phase', or else must be able to multiplex the different phases together to perform the correct updates. This difficult is compounded as the weight updates require the presence of both the gradient and the original activation simultaneously, thus necessitating the storage of the original activation during the relaxation phase. One potential solution to the multiplexing problem and the related problem of storing the original activations is through the use of multicompartment models of neurons with segregated dendrites. A number of algorithms have approximated backprop by using apical dendrites to keep separate representations of the activity and the error (Sacramento et al., 2018; Urbanczik & Senn, 2014). The apical dendrites of pyramidal neurons have many useful properties for this role – they receive a substantial amount of top-down feedback from other cortical and thalamic areas (Ohno et al., 2012), are electronically distant from the soma (Larkum, 2013), and they can operate as a 'third factor' in synaptic plasticity through NMDA spiking (Körding & König, 2001). Another potential solution to the multiplexing problem is that different phases could be coordinated by oscillatory rhythms of activity such as the alpha or gamma bands (Buzsaki, 2006). These oscillations could potentially multiplex the different phases together while handling continuously varying stimuli.

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

## APPENDIX A: ENERGY FUNCTION, EXTENSION TO ARBITRARY DAGS.

In this appendix, we elaborate on the mathematical background of the AR algorithm. We present a.) A candidate energy function that the dynamics appear to optimize, and discuss its limitations and b.) An extension of the AR algorithm to arbitrary computation graphs, which allow, in theory, for the AR algorithm to be used to perform automatic differentiation and optimisation along arbitrary programs, including modern machine learning methods.

### THE ENERGY FUNCTION

We first define the implicit energy function $\mathcal{E}$ that the dynamics can be thought of as optimizing. Given a hierarchical MLP structure with activations $x^l$, and activation function $f$ and parameters $\theta_i$ at each layer. This energy function is:

$$\mathcal{E} = \sum_{i=0}^{L} \frac{1}{2}x^{l2} - \frac{1}{2}f(W^l, x^l)^2 \tag{9}$$

Given this energy, function we can derive the dynamics as a gradient descent on $\mathcal{E}$

$$\frac{dx^l}{dt} = -\frac{\partial \mathcal{E}}{\partial x^l} = -x^l + f(W^l x^l)\frac{\partial f(W^l, x^l)}{\partial x^l} \tag{10}$$

$$= -x^l + x^{l+1}\frac{\partial f(W^l, x^l)}{\partial x^l} \tag{11}$$

where the second step uses the fact that in the forward pass $x^{l+1} = f(W^l, x^l)$. While this energy function can derive the required dynamics, it possesses several limitations. At the initial forward pass, the energy is 0 for all layers except potentially the output layer, and moreover it does not define a standard loss function (such as the mean-squared error) at the output. Thus the AR algorithm requires a special case to deal with the final output layer, as the 'natural' final layer loss is simply $T^2 - f(W^L, x^L)^2$ which is not a standard loss. Secondly, the value of the loss is not bounded above or below. For instance, it cannot easily be interpreted as a (squared) prediction error, which has a minimum at 0. Minimizing the energy function thus effectively requires trying to push $f(W^l, x^l)$ to be greater than $x^l$ as much as possible, rather than trying to equilibrate the two. Finally, the replacement of $f(W^l, x^l)$ with $x^{l+1}$ is a subtle issue due to the way the AR algorithm treats the activations in two separate ways. During the forward pass, this substitution is valid, however during the backward relaxation, the activations are changed away from their initial forward-pass values, and thus this substitution is not valid here.

EXTENSION TO DAGS

Here we consider the extension of the AR scheme described in the paper to more complex architectures. Specifically, we are interested in any function which can be expressed as a *computation graph* of elementary differentiable functions. This class includes essentially all modern machine learning architectures. A computation graph represents a complex function (such as the forward pass of a complex NN architecture like a transformer (Vaswani et al., 2017), or an LSTM (Hochreiter & Schmidhuber, 1997) as a graph of simpler functions [7]. Each function corresponds to a vertex of the graph while there is a directed edge between two vertices if the parent vertex is an argument to the function represented by the child vertex. Because we only study finite feedforward architectures (and since it is assumed finite we can 'unroll' any recurrent network into a long feedforward graph), we can represent any machine learning architecture as a directed acyclic graph (DAG). Automatic Differentiation (AD) techniques, which are at the heart of modern machine learning (Griewank et al., 1989; Van Merriënboer et al., 2018; Margossian, 2019), can then be used to compute gradients with respect to the parameters of any almost arbitrarily complex architecture automatically. This allows machine-learning practitioners to derive models which encode complex inductive biases about the structure of the problem domain, without having to be manually derive the expressions for the derivatives required to train the models. Here, we show that the AR algorithm can also be used to compute these derivatives along arbitrary DAGs, using only local information in the dynamics, and requiring only the knowledge of the inter-layer derivatives.

Core to AD is the multivariate chain-rule of calculus. Given a node $x^l$ on a DAG, the derivative with respect to some final output of the graph can be computed recursively with the relation

$$\frac{\partial L}{\partial x^i} = \sum_{x^j \in Chi(x^j)} \frac{\partial L}{\partial x^j} \frac{\partial x^j}{\partial x^i} \tag{12}$$

Where $Chi(x^i)$ represents all the nodes which are children of $x^i$. In effect, this recursive rule states that the derivative with respect to the loss of a point is equal to the sum of the derivatives coming from all paths from that node to the output. We now derive the AR energy function and dynamical update rule on a DAG. Specifically, we have in the forward pass that $x^i = f(W^j, x^j) \in Par(x^i))$ (where $Par(x)$ denotes the parents of x) so that each activation is some function of its parent nodes. We can thus write out the energy function and dynamics, and compute the equilibrium to obtain:

$$\mathcal{E} = \sum_{i=0}^{L} \frac{1}{2} x_i^2 - \frac{1}{2} f(W^j, x^j) \in Par(x^i))^2$$

$$\frac{dx^i}{dt} = -\frac{\partial \mathcal{E}}{\partial x^i} = x^i - \sum_{x^j \in Chi(x^i)} f(W^i, x^i) \in Par(x^j)) \frac{\partial f(W^i, x^i) \in Par(x^j))}{\partial x^i}$$

$$= x^i - \sum_{x^j \in Chi(x^i)} x^j \frac{\partial x^j}{\partial x^i}$$

$$\frac{dx^i}{dt} = 0 \implies x^{i*} = \sum_{x^j \in Chi(x^i)} x^{j*} \frac{\partial x^j}{\partial x^i} \tag{13}$$

Crucially, this recursion satisfies the same relationship as the multivariable chain-rule (Equation 11), and thus if the output nodes are equal to the gradient at the top-level such that $x^L = \frac{\partial L}{\partial x^L}$, at equilibrium the correct gradients will be computed at every node in the graph. Thus AR can be converted into a general-purpose AD algorithm which utilises only local information. Neurally, this only requires the dynamics in the relaxation phase to respond to the sum of top-down input, which is very plausible, and thus perhaps suggests that it is possible the brain could be utilizing substantially more complex architectures than feedforward MLPs.

---

[7]Although AR can be applied to any DAG, including the unrolled graphs of recurrent networks like LSTMs, we do not claim that this applying AR in this way is biologically plausible, since such an approach would require propagating the AR update rules backwards through time.

## APPENDIX B: CROSS-ENTROPY LOSS FUNCTION

It is important to note that although we use mean-square-error (MSE) in our main experiments, AR does not depend on any specific loss function. All that is necessary to use a different loss function is clamp the activations of the final layer to the derivative of the chosen loss function during the relaxation phase. With these units clamped correctly, the AR algorithm will be able to converge to the correct backpropagated gradients for any loss function. To demonstrate this we also present results with the multiclass cross-entropy loss function $L = \sum_i T_i, x_i^L$. For these experiments we used the same neural network architecture in the main paper except that the activation function at the output layer was a softmax (instead of linear), to ensure that the outputs correspond to valid probabilities, as required for the cross-entropy loss.

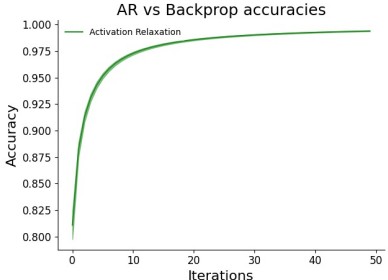

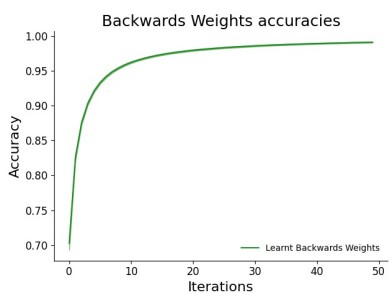

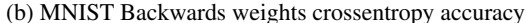

(a) MNOST standard AR crossentropy accuracy

(b) MNIST Backwards weights crossentropy accuracy

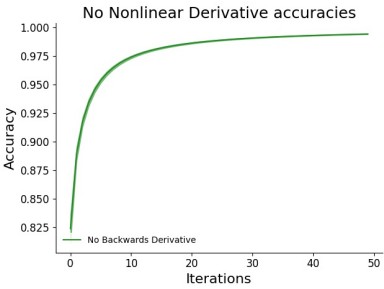

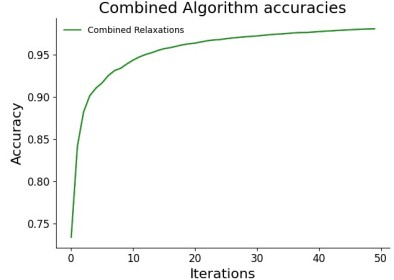

(c) MNIST no nonlinear derivatives crossentropy

(d) FMNIST combined AR crossentropy accuracy

Figure 5: Accuracy (averaged over 5 seeds) for AR trained with the crossentropy loss function in the standard AR, learnable backwards weights, no-nonlinearities, and combined conditions.

## APPENDIX C: SCALABILITY TO CNNs

We present empirical results showing that AR can scale to larger, more complex network architectures and more challenging tasks than MNIST and Fashion-MNIST. Here we apply AR to a small CNN architecture, and test for classification accuracy on the harder CIFAR dataset, composed of 10 classes of 32x32x3 images. The input pixel values were normalized to lie between 0 and 1. The CNN comprised two convolutional layers with maxpooling, following by two fully connected layers. The convolutional layers used 32 and 64 convolutional filters while the fully connected layers had 64, 120, and 10 neurons respectively. All layers used the hyperbolic tangent nonlinearity, except the final layer which was linear. The network was trained using the mean-square-error loss function.

We showcase performance against the backprop baseline of our CNN under the three simplification conditions of our algorithms – with learnable backwards weights, with no nonlinear derivatives, and with both simplifications combined. The AR CNN performs comparably with backprop even with these simplifications applied, thus providing preliminary evidence of the scalability of the approach. For more information on the scalability of this algorithm, please see (Millidge et al., 2020b). In

general, we believe that investigating the potential scalability of these algorithms to truly challenging architectures and tasks such as imagenet is an exciting area for future research.

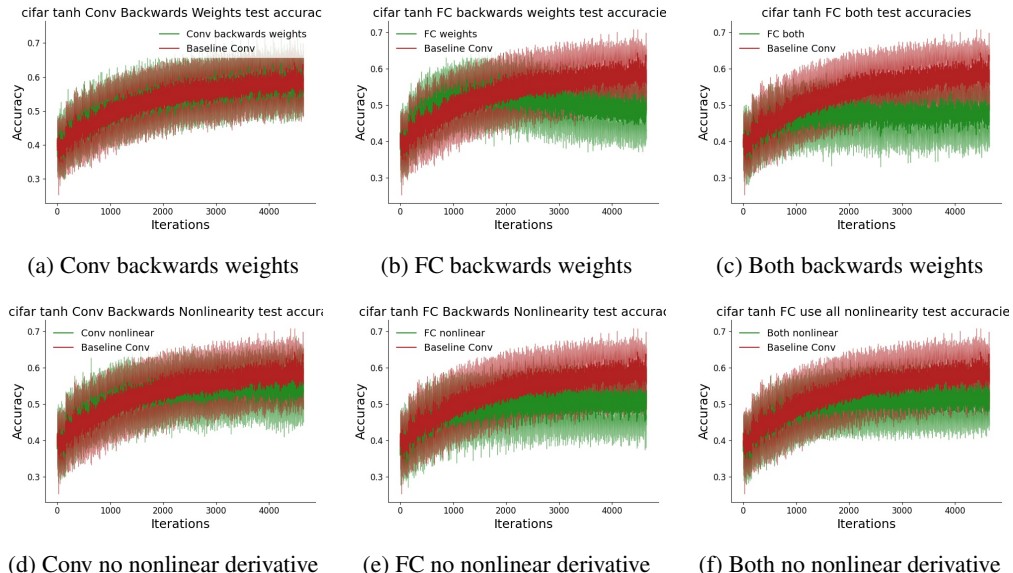

(a) Conv backwards weights  (b) FC backwards weights  (c) Both backwards weights

(d) Conv no nonlinear derivative  (e) FC no nonlinear derivative  (f) Both no nonlinear derivative

Figure 6: Performance (test accuracy), averaged over 10 seeds, on CIFAR10 demonstrating the scalability of the learnable backwards weights and dropping the nonlinear derivatives in a CNN architecture, compared to baseline AR without simplifications. Performance is equivalent throughout.

## APPENDIX D: HYPERPARAMETER TABLE FOR MAIN EXPERIMENTS

All hyperparameter values were selected after preliminary experimentation but without a large-scale grid search. We typically used standard hyperparameter values taken from the deep learning literature on training ANNs, and found that they worked reasonably well with AR. The most important AR-specific hyperparameter is the learning rate $\eta_x$ for the relaxation phase. We undertook a hyperparameter sweep for this parameter, as shown in Figure 1 (right), and settled on a value of 0.1 which provided a good balance between rapid convergence and numerical stability.

| Hyperparameter | Value |
|---|---|
| Batch Size | 64 |
| $\eta_\theta$ | 0.0005 |
| Relaxation Iterations | 100 |
| $\eta_x$ | 0.1 |
| Weight Initialization Variance | 0.05 |

