# OpenReview forum: "Activation Relaxation: A Local Dynamical Approximation to Backpropagation in the Brain"
_ICLR.cc/2021/Conference — Reject_

### Official Review · AnonReviewer1 · 2020-10-24
**Major point needs attention from the authors**

**Rating:** 4
**Confidence:** 4

**Review:**

The authors propose a biological model for loss function differentiation which reuses a single pool of neurons to compute predictions and prediction errors (i.e., derivatives of a loss function with respect to the neural acitivity variables) in two sequential phases.

I have a major concern that I need to see addressed by the authors:
- In my understanding the proposed method is not local in time and this is not at all clear, the way the paper is written. This is best seen on the last line of Algorithm 1: the apparently innocuous term $\frac{\partial x^l}{\partial W^l}$ should be evaluated at the _feedforward activation values_, not using the equilibrium $x^l$. I wonder whether the authors are aware of this, as such a detail could be somewhat easily overlooked when using automatic differentiation.

Note that this is very different from dropping $f^\prime$ (as per footnote 5) from the gradient, something that has been experimented with in some of the feedback alignment implementations in the past (for example, some of the simulations in Lillicrap et al., 2016, do not backpropagate $f^\prime$, although the local ${f^\prime}^l$ is still used when updating $W^l$).

Stemming from this issue, the advantages of the proposed scheme against standard predictive coding implementations using two populations of neurons (predictions and errors; e.g., Whittington & Bogacz, 2017) aren't clear. As I understand the algorithm, prediction activity is overridden by error activity (in a way that still requires coordination/phases), but since prediction activity is necessary to compute the aforementioned term, the resulting learning rule becomes nonlocal in time.

A proper evaluation of the significance of the advance brought by the method requires clarifying this issue.

I leave some additional comments for the authors:
- "Given that backprop provides an optimal solution to this problem (Baldi & Sadowski, 2016)"
The claim that backprop provides an optimal solution to the credit assignment problem is a very strong one, and I'm not sure that it is supported by the cited reference. Optimal gradient approximation is different from optimal credit assignment.
- "Unlike contrastive Hebbian methods, AR does not require the coordination and storage of information across multiple backwards phases, which would pose a substantial challenge for decentralised neural circuitry."
While AR does not require storage of information across multiple backwards phases, it requires storage of information across forward and backward phases.This is related to my major comment above, and this issue/potential misunderstanding reappears in multiple points of the paper.
- In (1), why is $x^L$ and not $x^l$ appearing?
- In the equation below (2), should $x_i$ be $x^l?
- Figure 3: these curves are rather hard to see, perhaps consider presenting mean +- error bars over many runs?
- I could not follow several of the steps in the arguments presented in the appendix titled "The energy function". How is (11) related to the dynamics of the algorithm?
- There are a few typos that should be corrected, e.g., "is designed to converges to the minimum".

---
Edit after rebuttal:

I am sorry for not yet being able to support acceptance of this paper. I appreciate the authors' work during the rebuttal, and I agree that the execution of the paper improved in this version.

But the issue remains that the proposed algorithm does not fundamentally lift any of the current obstacles in implementing backpropagation in the brain. Most importantly, the algorithm is very similar to exact standard backpropagation, and it still requires the same coordinated phases as backpropagation: the forward phase and the backward phase. The learning rule is non-local in time and while it does not use activity differences, that is not necessarily a good thing. For example, the connection to spike-timing-dependent plasticity becomes harder to establish. The authors verify that learning still works after important approximations are made (notably to avoid weight transport), but these approximations were already previously reported and are more or less obviously applicable here, given how close activation relaxation is to standard backpropagation.

Finally, I really recommend the authors to more explicitly state and discuss the temporal non-locality of the algorithm when introducing the method (and in Algorithm 1), which remains insufficiently clear to me.

---

> ### Author Response · Authors · 2020-11-22
> **response to reviewer 4**
>
> We thank the reviewer for their insightful comments. We indeed agree that the dx^{l+1}/dW term must be evaluated at the feedforward pass values. We discuss this in the discussion section, and (we believe) make it clear in the methods with the vert-bar notation. If this is not clear in the manuscript, we would welcome suggestions as to how to make this more clear since, as the reviewer notes, it is an important point.
>
> This term does in fact require that the AR algorithm must store information between the forwards and backwards passes, which is a limitation on its biological plausibility. In future work we aim to further develop alternatives to this algorithm which can work around this limitation.
>
> We agree that this limitation makes its advantage over other methods such as predictive coding and equilibrium-propagation less clear, since all such methods require storing information from the forward phase (or "free phase" in EP) for use in computing gradients in the backwards phase (or "clamped phase" for EP). This seems to be a general property of recurrent algorithms which can converge to the exact backprop gradients, which makes sense since the backprop gradients fundamentally depend on the feedforward activations. The advantage of EP over these other methods is really the simplicity of its update rules and not requiring multiple populations of neurons as in predictive coding. We also believe it is valuable to have a record of all the possible algorithms for approximating backprop, so we can gain a general view of the overall landscape.
>
> You make a good point about the optimality of backprop for credit assignment -- we have reworded this to be less strong
>
> We have also fixed the typos mentioned in your review -- thanks for these.
>
> Regarding your point about the energy function. The idea is that the dynamics can be derived as a gradient descent on the energy function -- that equation 5 can be derived as a gradient descent on equation 9. As is shown in equations 10 and 11. Does this clarify things? We are happy to go into more detail about this if there is any confusion still remaining.

---

### Official Review · AnonReviewer3 · 2020-10-28
**Interesting idea, few major concerns**

**Rating:** 7
**Confidence:** 4

**Review:**

## Second Review

Thanks for taking all my comments seriously.  After carefully reading other reviews and quick modifications introduced by authors, I believe this work is richer and has shown some potential towards building scalable and robust alternative to BP.  Thanks for including angles as suggested, this further supports  hypothesis proposed in this work. I really appreciate results using CNNs and cross entropy loss, therefore I increase my score to 7. It seems that other reviewers do not appreciate that training a network using hebbian like updates and without  BP requires some nontrivial engineering tricks and theoretical considerations which are now well described in this paper (updated manuscript). It is difficult to match BP gradients, and many popular alternatives including FA, DFA, DTP, EP struggle to match BP performance when tested on complex benchmarks (Cifar, imagenet, etc) with complex architectures (CNNs, RNNs, etc). Beside this given approach is robust even when backward weights are trainable. However I agree storing backward and forward synapses challenges the bio-plausibility of approach, which I think can be handled if local representation are handled differently. Nonetheless, changing few strong words (optimal BP, optimal gradients) and derivation, supports major hypothesis proposed in this work. A better justification on non-local updates as raised by other reviewers is required to further strengthen this work. But i liked the results with activity relaxation and how close gradients are with respect to BP. Combining current approach with other bio-inspired approaches might solve some key aspects of current learning algorithm.
Current approach is still heavily dependent on backprop, and partially gets closer to bioplausible approaches (mainly hebbian like update rule). Testing this on deeper architectures (Resnet, student-teacher etc) might further strengthen your work.

## Minor comments

Figure 4 c) change angls--> angles
In appendix, change caption of fig 5a) from MNOST to MNIST, I believe it is a typo

Add results with FA or DFA with fixed and learnable weights, will further support robustness and closeness of BP claim in this work.

## Summary
In this work, authors designed a bio-inspired approach known as activation relaxation (AR), utilizing local information for training deep neural network. Unlike prior work on bio-inspired learning, AR only utilizes single neuron to compute its gradient helpful for neural circuitry. AR is seen to be derived from postulating a dynamical relaxation phase in which the neural activities are tracing out a dynamical system. This modification is hypothesis to be close to backprop gradients getting ride of weight transport problem. Authors show accuracy as a metric to validate their hypothesis by conducting experiments on small scale dataset such as mnist and fmnist.

## Review
Most of the paper is clear, but experiment section needs more work in approving the hypothesis w.r.t AR.

1] AR converges close to backprop gradients.
Accuracy is not a valid measure to show robustness and calculate approximately with backprop gradients.  As shown by FA (lillicrap) , DFA (nokland), LRA (ororbia), the updates carried by there model roughly lies within 45 degree compared with backprop. It is better to show model update angles w.r.t to BP, DFA and other bio-inspired approaches.

2] is AR robust? What happens to the model at various initialization and how does it behave when tested with various model hyper-parameter changes?

## Experimental section is missing major chunk of information.
2.1] Did you perform grid search on model hyper-parameters, if so, what are those? Did you experiment with various activation functions and if so, what are those? Are results consistent whenever MSE is replaced with CE? If so report your findings.
2.2] What happens to the network when BP gradients are weak resulting into poor information, can AR come out of that low saddle point and converge to better local minima? Few experiments related to better convergence are shown by ororbia, they experimented with various init to show their model can converge to better local minima whenever backprop had issue in converging.
2.3] Did you perform multiple trails on your experiments? If so, you should report standard error in your paper.

3] AR w.r.t fixed error weights vs AR w.r.t learnable error weights.
How does local learning approach, help in improving the model plasticity when you have these two scenarios? Paper mentions few lines on those, but detailed experiments should be conducted to validate the statement stable and robust performance. Also provide information on how error weights are initialized, and ranges experimented w.r.t backward or error weights.

4] Comparison with other bio-inspired approaches.
Current manuscript does not show any comparison with other bio-inspired approaches (DFA, FA, DTP, DTP-sigma, LRA, Weight mirroring). If goal is to show model is robust, close to backprop updates and gradients, then it is better to show comparison with these approaches or at least show angle to understand closeness w.r.t BP updates.

5]  Performance on large scale dataset and CNNs.
It is been argued that bio-inspired approaches (DFA, FA, DTP) struggle to match backprop performance when evaluated on large scale dataset with deep CNNs[Bartunov 18] . However recently weight mirroring [Akrout 19] and LRA[Ororbia and Mali 19]  have shown that they can come closer to backprop in terms of performance on large scale datasets. But in this current manuscript, there are no results w.r.t CNNs and updates w.r.t. filters when AR is deployed on such challenging visual recognition tasks.

[Bartunov 18] Bartunov, S., Santoro, A., Richards, B., Marris, L., Hinton, G.E. and Lillicrap, T., 2018. Assessing the scalability of biologically-motivated deep learning algorithms and architectures. In Advances in Neural Information Processing Systems (pp. 9368-9378).

[Ororbia 18] Ororbia, A.G., Mali, A., Kifer, D. and Giles, C.L., 2018. Deep credit assignment by aligning local representations. arXiv preprint arXiv:1803.01834.

[Akrout 19] Akrout, M., Wilson, C., Humphreys, P., Lillicrap, T. and Tweed, D.B., 2019. Deep learning without weight transport. In Advances in neural information processing systems (pp. 976-984).

[Ororbia and Mali 20] Ororbia, A., Mali, A., Kifer, D. and Giles, C.L., 2020. Reducing the Computational Burden of Deep Learning with Recursive Local Representation Alignment. arXiv preprint arXiv:2002.03911.

---

> ### Author Response · Authors · 2020-11-22
> **response to reviewer 3**
>
> We thank the reviewer for their very detailed and insightful review. These suggestions for additional experiments will definitely help in improving the manuscript.
>
> The reviewer has raised several interesting suggestions which we have implemented to improve our results. We have substantially changed the results section and added considerable more detail about the experiments undertaken.
>
> Firstly, we agree, that an important way to understand the results of AR is to directly compare the weight angles. We have added plots of the angle throughout training both for the standard AR algorithm as well as the simplifications in the second section of the paper. Interestingly, the gradient angle computed by the standard AR algorithm typically lies within 10-15 degrees of the backprop updates, thus performing better than FA and DFA in this regard, which we believe is reflected in the closer match to backprop. Interestingly, in the case of learnable backwards weights, we see an initial starting very large weight angle (as in the FA case) which then declines over time as the backwards weights are learnt -- thus validating that the backwards weights are being learned correctly.
>
> Secondly, we have described in more detail the initialization of the weights including the learnable backwards weights and included a full table of hyperparameter values in an appendix. We also ran each experiment for five seeds (which we did before but this was not visible due to the noise of plotting each minibatch) and plotted error bars of the standard error between runs.
>
> We are currently working on demonstrating AR with a different (crossentropy) loss function as suggested. Experiments for this are running and will hopefully be ready by tomorrow.
>
> Finally, we have included preliminary results showing that AR, including with the learnable backwards weights and dropped nonlinearities can scale up to more challenging tasks such as training CNN architectures on the CIFAR10 and CIFAR100 datasets.
>
> We agree with the reviewer that these additions to the results are necessary and we hope that we have substantially improved the paper and the testing of our hypothesis.

---

### Official Review · AnonReviewer4 · 2020-10-28
**Biologically plausible algorithm for backprop**

**Rating:** 8
**Confidence:** 2

**Review:**

The authors propose an algorithm for estimating the correct backprop gradients that is described as biologically plausible. I have very little to contribute as I think this is a clearly-written paper — as such I recommend acceptance.

I would be happier if the authors gave a very very strict definition of the kind of plausibility they wish to capture with falsifiability criteria and deep theoretical underpinnings, as I think it's a bit of a semantic trap without reference to some explicit level of analysis, etc. But this is a field-wise terminological issue, and thus can easily be seen as out-of-scope.

Minor: A paper the authors might appreciate is, given some of the work they mention to motivate their neuro/bio plausible claims:

Xu, Y., & Vaziri-Pashkam, M. (2020). Limited correspondence in visual representation between the human brain and convolutional neural networks. bioRxiv.
https://doi.org/10.1101/2020.03.12.989376

---

> ### Author Response · Authors · 2020-11-22
> **response to reviewer 2**
>
> We thank the reviewer for this insightful and positive review. We agree about the vagueness of the term biological plausibility, and have added a paragraph in the introduction of the revised version of the paper explaining this term in more depth:
>
> ------
>  It is important to note that biological-plausibility is an amorphous term in the literature, with many possible readings. Here, we specifically mean that the algorithm requires only local information to be present at each synapse, and information is processed according to straightforward linear, or Hebbian-like update rules. It is important to note that here (as in much of the literature), we use the abstraction of rate-coded leaky-integrate-and-fire neurons. The extension to more neurophysiologically grounded spiking models is an exciting area of future work.

---

### Official Review · AnonReviewer2 · 2020-10-29
**Nice but not sufficient**

**Rating:** 4
**Confidence:** 4

**Review:**

## Summary

The authors suggest a "local" algorithm to replace backpropagation in feedforward neural networks. Locality is meant here by the compatibility with the NGRAD hypothesis: backpropagation through the layers is replaced by the propagation of the network activity itself. Therefore there is no need for a second implicit error network or error neurons.

This method introduces dynamics in the neuron to converge towards an equilibrium and remark that, at equilibrium, the activity transmitted backward becomes equal to the back-propagated gradients. The method is tested on MNIST and fashion MNIST.

## Critical review

The methods seems extremely similar to equilibrium propagation and the account of this resemblance does not seem to be acknowledged in a fair manner. Rather the introduction summarized a large spectrum of plausible learning rules which are more or less related to AR and the authors do not comment in a clear way about the technical resemblance with these algorithms.

The derivation originate from an interesting observation at equilibrium, but this is certainly not as rigorous as the proofs given for equilibrium propagation. Equilibrium propagation has been demonstrated to work on tasks harder than MNIST and fashion MNIST.

The simulations are not sufficient to show if the method is competitive or practicable. The plots show the learning curves where the test accuracy is spanning values between 92.5 and 98 on MNIST. Given the variance, this is probably computed only on mini batches of the test set and not over the entire test set. Hence, it is not the regular protocol for reporting the test accuracy. A good back-prop implementation is expected to reach 98.2% accuracy on the test set on MNIST. I do not know whether the BP baseline reaches this level in this paper. It vaguely seem that AR is similar to their BP implementation but I do not know what to judge from this and the simulation results and not commenting in details in the text.

I strongly doubt the statement that the method applies to recurrent networks and LSTMs, it may work for any DAG, but in a recurrent network the temporal dynamics are already meant to model the network computation and the changing inputs, therefore the network cannot simply converge to an equilibrium in this case. Also the activity in NGRAD replace BP by transmitting activity backward through the depth but this does not make sense to replace back-propagation though time in recurrent networks since biological neural cannot transmit activity backward in time.

As a conclusion, I see a few interesting remark in the paper but neither the simulation nor the theory are bringing substantial insights to the plausible learning rule community or the broader representation learning literature.

I did not undertand what is meant by "the Jacobian is negative-definite".

---

> ### Author Response · Authors · 2020-11-22
> **Response to reviewer 1**
>
> We thank the reviewer for their insightful and helpful review, which will undoubtedly help improve the manuscript.
>
> Regarding equilibrium-prop, our method is substantially different from equilibrium-prop (EP). Our method only requires one dynamical relaxation phase instead of the two (free phase and clamped phase in EP), and the gradients are represented in the activities of the neurons at the end of the activation phase, rather than the difference in activities between phases (as in EP). Additionally the update rules in the relaxation phase differ from the update rules of EP. Finally, the energy functions the update rules are derived from are different. You raise a good point though about the lack of specific comparisons. We have added detailed comparisons of our method to three major alternatives (EP, predictive coding, and local representation alignment (LRA)) in the related works section.
>
> Could you perhaps be more specific on how we could improve the rigour of our derivations of the method. We present the results at equilibrium, but also show that the update rules are guaranteed to converge to equilibrium. Convergence is guaranteed due to the fact that the dx^l+1/dx^l derivative is computed at the feedforward pass values, which is constant during the relaxation phase. This turns equation 5 into a linear ODE which is guaranteed to converge to the fixed point. We also verify this numerically.
>
> We thank the reviewer for their helpful suggestions on how to improve the results section. We agree that the initially presented results were not sufficient to demonstrate the hypothesis. Taking your suggestion, we have redone the results to follow the more standard approach of averaging the training and test accuracy over epochs, generating smoother curve. We have also computed and commented upon the angle between the AR-computed gradients and the true backprop ones for the standard AR method, and also the simplifications introduced in part II. We hope these additional results will aid demonstrating the validity of the method.
>
> AR can directly be applied on recurrent networks such as LSTMs and RNNs since these networks (assuming they are run in finite time) can be unrolled across time to form a DAG computation graph. We agree with the reviewer, however, that this does not attempt to approximate BPTT in neural systems (which does, as noted, require propagating the AR update rules backwards through time. We have added a footnote to this effect in the appendix where the extension is discussed.

---

### Decision · Program_Chairs · 2021-01-07
**Final Decision**

**Decision:**

Reject

**Comment:**

The authors propose an algorithm to perform backprop in a feed-forward neural network without the need to backpropagate errors. They hence claim that this algorithm is a biologically plausible variant of Backprop.

After a forward-propagation phase, the method introduces a relaxation phase and they remark that at the equilibrium of this phase, the activity is equal to the derivatives. Some related algorithms have been proposed previously (predictive coding, equilibrium-prop, target-propagation). Advantages of the proposed algorithm are that it does not need multiple distinct backwards phases and that it only utilizes a single type of neuron instead of separate populiations (such as in predictive coding).

Their method is tested on MNIST and fashion MNIST using a 4-layer fully connected network. In a revised version after the initial reviews, the authors added preliminary results on CIFAR-10 with a 4 layer CNN.

The authors then study the impact of some unbiological constraints such as symmetric weights.

While the reviewers agreed that the work is interesting, there was some disagreement on the significance of the model. In particular, it was noted that while the learning rules are indeed local in space, they are not local in time (the network has to remember variables from the forward phase until the update at the relaxation phase), which was deemed questionable from the biological perspective.

In addition, it was criticized that the simulations are no sufficient to support the claims of the paper. The datasets are relatively simple, networks are shallow and performances of baseline models are not state-of-the-art.
 In a revised version after the initial reviews, the authors added preliminary results on CIFAR-10 with a 4 layer CNN. However, these results do not seem conclusive, as the baselines are far below SOTA and networks are still quite shallow (a study by Lillicrap found that biological approximations to backprop struggle especially when applied to deep networks).

In summary, the work looks promising but some questions remain about the locality of learning and its applicability to more demanding tasks.

I add that one reviewer gave a very good rating with a poor review and did not respond to any questions about the justification. Therefore I had to neglect this review.